# Pri-miR526b and Pri-miR655 Are Potential Blood Biomarkers for Breast Cancer

**DOI:** 10.3390/cancers13153838

**Published:** 2021-07-30

**Authors:** Mousumi Majumder, Kingsley Chukwunonso Ugwuagbo, Sujit Maiti, Peeyush K Lala, Muriel Brackstone

**Affiliations:** 1Department of Biology, Brandon University, Brandon, MB R7A 6A9, Canada; ugwuagkc97@brandonu.ca (K.C.U.); maitis@brandonu.ca (S.M.); 2Department of Oncology, Schulich School of Medicine and Dentistry, University of Western Ontario, London, ON N6A 5W9, Canada; pklala@uwo.ca (P.K.L.); muriel.brackstone@lhsc.on.ca (M.B.); 3Department of Anatomy and Cell Biology, Schulich School of Medicine and Dentistry, University of Western Ontario, London, ON N6A 3K7, Canada; 4Lawson Research Institute, London, ON N6C 2R5, Canada

**Keywords:** pri-miRNA, miRNA, biomarker, breast cancer, early detection, plasma

## Abstract

**Simple Summary:**

Previously, we reported that the expression of two oncogenic miRNAs, miR526b and miR655, in poorly metastatic breast cancer cells enhances aggressive breast cancer phenotypes. Furthermore, miR526b and miR655 expression in breast tumors is associated with poor patient survival. We recently showed that both miRNAs are major regulators of the tumor microenvironment and can be detected in cell-free tumor cell secretions. Therefore, we wanted to test the biomarker potential of these two miRNAs. Early detection can improve breast cancer patient survival by 98%. Here, we report novel findings that precursors of both miRNAs, pri-miR526b and pri-miR655, are sensitive and robust blood biomarkers to distinguish cancer from benign plasmas. Pri-miR526b proved to be a very sensitive biomarker in detecting breast cancer at an early stage. Hence, pri-miR526b can be used as an early diagnostic biomarker for breast cancer.

**Abstract:**

We reported that two microRNAs, miR526b and miR655, are oncogenic in breast cancer (BC). Overexpression of these two miRNAs in poorly metastatic BC cells promotes aggressive BC phenotypes in vitro and in vivo. High expression of each miRNA was associated with poor patient survival. In this pilot biomarker study, we report for the first time that miRNA precursor RNAs (pri-miRNAs) are robust and sensitive biomarkers for BC, detectable in both human blood plasma and biopsy tissues. Pri-miRNA detection and quantification do not require a special enrichment procedure, thus reducing specimen quantity. Blood plasma samples from 90 malignant tumor-bearing patients and 20 benign lesion-bearing participants (control) were analyzed for pri-miRNA expression with a quantitative real-time polymerase chain reaction. Results revealed that normalized expressions of plasma pri-miR526b and pri-miR655 are significantly upregulated in malignancy compared to benign plasmas (*p* = 0.002 and *p* = 0.03, respectively). Both pri-miRNAs showed more prominent results to distinguish stage I plasmas from benign plasmas (*p* = 0.001 for pri-miR526b and *p* = 0.0001 for pri-miR655). We have also validated pri-miRNA expression in independent tumor bank tissues, showing significant upregulation of both pri-miRNAs in BC; thus, pri-miRNAs are robust markers. The diagnostic relevance of pri-miRNAs was computed with the area under the curve (AUC). Pri-miR526b is a sensitive biomarker to distinguish cancer from control plasmas (sensitivity of 86%; AUC = 71.47%, *p* = 0.0027) with a positive predictive value of 88.89%; however, pri-miR655 did not show significant sensitivity. Furthermore, pri-miR526b could also significantly distinguish tumors as early as stage I from control (sensitivity of 75%; AUC = 72.71%, *p* = 0.0037). Therefore, pri-miR526b can be used as an early diagnostic biomarker. The expression of both pri-miRNAs was significantly high in ER-positive and HER2-negative subgroups of BC; hence, these biomarkers might play a role in the management of endocrine therapy designs. Additionally, with a case–control cohort study, we identified that high expression of pri-miR526b in the blood is also a risk factor associated with breast cancer (OR = 4.3, CI = 1.39–13.34, *p* = 0.01). Pri-miRNAs could be considered novel breast cancer blood biomarkers.

## 1. Introduction

Breast cancer (BC) is the most common solid organ-specific cancer, affecting about 30–40% of women under the age of 40 years in North America [1]. While BC accounts for about 15% of all cancer-related deaths in women, early detection and treatment strategies have contributed significantly to reducing disease-related mortality [2,3]. Mammographic screening is a painful breast examination procedure, currently used as the gold standard for tumor detection. However, this screening excludes women below the age of 50 years in Canada [4] and is limited by a high percentage of false positive results, which requires further investigation for molecular signatures using invasive biopsy techniques [5]. A blood test can be a less invasive procedure for BC screening, and there are a few routine cancer markers in the blood, such as carcinoembryonic antigen (CEA) and carbohydrate antigen (CA)15-3, which have been used as biomarkers. Their low sensitivity and specificity to detect disease make them poorly reliable screening tools [6]. Hence, breast cancer screening requires a sensitive blood biomarker. Recently, the identification of microRNAs (miRNAs) in body fluids and specifically in the blood make them strong candidates as cancer biomarkers [7].

MiRNAs are endogenous noncoding small RNA (22 nt) molecules that regulate gene expression at the post-transcriptional level. Primary microRNAs or “pri-miRNAs”, which can be more than 1000 nt in length, contain an RNA hairpin in which one of the two strands includes the mature miRNA. The hairpin, which typically comprises 60–120 nt, is cleaved from the pri-miRNA in the nucleus by the double-strand-specific ribonuclease, Drosha. The resulting precursor miRNA, or “pre-miRNA”, is transported from the nucleus to the cytoplasm, where it matures to become miRNA [8].

MiRNAs, and to a lesser extent pri-miRNAs and pre-miRNAs, are secreted by both tumor and healthy cells into the interstitial fluid contributing to the fluid part of the tumor microenvironment (TME) [9]. Circulating miRNAs are exported by exosomes into body fluids, such as blood plasma, and they have recently emerged as candidate biomarkers for detecting and monitoring disease progression in cancer patients [10]. Only a few mature miRNAs in the blood plasma of BC patients have shown promise for the detection of malignancy [11,12]. On the other hand, the diagnostic usefulness of circulating double-stranded RNA-like pre-miRNA or pri-miRNA is a very new field. The first report on plasma pri-miRNA as a cancer biomarker was published for lung cancer [13]. However, very few reports are available for pri-miRNAs as a biomarker of cancer, likely because of their low abundance in the plasma.

We have shown that overexpression of cyclo-oxygenase (COX)-2 promotes breast cancer progression and metastasis via multiple mechanisms [14,15], including induction of two oncogenic miRNAs miR526b [16] and miR655 [17]. Overexpression of these two miRNAs in poorly metastatic cell lines promoted cancer cell migration, invasiveness, and stem-like cell phenotypes; moreover, upon orthotopic transplantation in immune-compromised mice, they enhanced tumor growth and metastasis. Furthermore, miR526b and miR655 overexpression in human breast cancer was correlated with reduced patient survival and disease progression [16,17]. However, the biomarker potentials of these two miRNAs or pri-miRNAs for the early diagnosis of breast cancer remain unexplored.

Here, we designed a pilot biomarker study using blood plasma of breast cancer-bearing patients and benign subjects with cancer-free benign breast lesions, to test the clinical usefulness of pri-miRNAs as blood biomarkers for early detection of breast cancer. We tested the expression of pri-miRNAs in various tumor stages and subtypes to use that information to monitor therapeutic progression and manage treatment regimens. We also conducted a case–control cohort study by testing overexpression of these pri-miRNAs as a risk factor for breast cancer.

## 2. Materials and Methods

### 2.1. Study Design, Patient Demography, and Study Subjects

The permission for this study was granted by Brandon University Ethical Approval Committee (approval #21968), as well as the Ethics Research Board of the University of Western Ontario (approval #105400). This study is a pilot biomarker study, consisting of plasma samples from diagnosed breast cancer patients (case) and participants with cancer-free benign diseases (control). A total of 110 blood samples (90 case and 20 control) were collected from participants with their written consent at the time of biopsy (or lumpectomy) for suspected breast cancer at the London Regional Cancer Program (LRCP), and specimens were preserved and maintained at the London Tumor Biobank (LTB). Matched plasma and biopsy tissue samples were then transferred to the University of Western Ontario, and aliquots of all samples were then transferred to Brandon University with a Material Transfer Agreement signed between both universities. Sample processing, RNA extraction, all experiments, and data analysis were conducted at Brandon University.

Study samples were from female subjects of middle to late age groups (42–91 years). Tumor samples from BC subjects were grouped according to TNM staging and ER, PR, and HER2 status. The study design was divided into two major steps: miRNA and pri-miRNA detection in blood plasma and validation. In a few plasma and matched biopsy samples, we verified the expression of both mature miRNAs and pri-miRNAs. Expressions of miR526b, miR655, pri-miR526b, and pri-miR655 were measured, and RNU44 and RPL5 were considered as reference markers for miRNA and pri-miRNA expression, respectively. We also conducted a case–control cohort study using the same sample sets to test pri-miRNA expression as a risk factor for breast cancer onset.

In addition, to test the robustness of pri-miRNA as a marker, we used *n* = 20 breast cancer and *n* = 20 disease-free control tissues collected from the Ontario Institute for Cancer Research (OICR) Tumor Bank and validated miRNA and pri-miRNA expression in these tissues. We processed OICR samples similarly for RNA and miRNA extraction and gene, miRNA, and pri-miRNA expressions mentioned below. However, no blood plasma sample was available in the OICR Tumor Bank.

### 2.2. Clinical Sample Preparation and Storage

The tissues were flash-frozen in dry ice immediately after surgery and stored in liquid nitrogen until further processed. A 2 mL volume of matched venous blood samples from the study subjects was collected in anticoagulant-containing tubes and spun; cell-free plasma was aliquoted into 2 mL RNAse-free cryovials within 10–30 min. All samples were preserved at −80 °C for storage.

### 2.3. RNA Extraction and cDNA Synthesis

Total RNA was extracted from plasma and biopsy tissues using an miRNeasy mini kit (Qiagen^®^, Toronto, ON, Canada). We used a standardized plasma miRNA extraction procedure with a starting volume of 350 µL of blood plasma, adapted from Qiagen purification protocol for mRNA and miRNA [18]. We used 5 mg of the flash-frozen biopsy tissues to extract both RNA and miRNA. This yielded at least 30 µL, with a total of 500–600 ng of total RNA, a fraction of which was further enriched for mature miRNA purification (200 ng miRNA).

cDNA was synthesized using a SuperScript cDNA reverse transcription kit (ThermoFisher Scientific, ON, Canada). A highly specific miRNA-specific cDNA reverse transcription requires at least 10 µL of miRNA (10–50 ng). Thus, each miRNA-specific cDNA synthesis would require a large number of patient specimens for screening. To resolve this challenge, we considered using pri-miRNA, which can be extracted using the regular RNA extraction column and cDNA synthesized from total RNA (containing large RNAs, small RNAs) using a random primer. We used 500 ng of total RNA to prepare cDNA.

### 2.4. Quantitative Real-Time PCR (qRT-PCR) 

The TaqMan MiRNA, Pri-miRNA, and Gene Expression Assay Kits (TaqMan Probes) were used for real-time quantitative PCR (ThermoFisher Scientific, ON, Canada). Both miRNA and pri-miRNA transcripts were quantified using the Ct values. Relative quantification of plasma miRNA/pri-miRNA in both BC and control subjects was calculated as delta Ct (∆Ct) = Ct of miRNA or pri-miRNA − Ct of endogenous control. Previously, we showed that RNU44 is a good control marker for miRNA expression and did not show variability in cell lines and breast tissues [16,17]. Similarly, we chose RPL5 as the control gene for gene expression as it showed consistent expression without a significant variability in various tissues and cell lines [19,20,21].

A negative ΔCt denotes high miRNA/pri-miRNA expression [22]. A normal distribution curve was conducted across the tumor and control samples distribution to define a cutoff of high- and low-expression samples. For each sample, duplicate runs for each probe were performed, and a total of 5% of the samples quantified again for all markers for the second time to confirm consistency of the pri-mRNA, miRNA, and gene expression.

### 2.5. Receiver Operating Characteristic (ROC) Curve

To evaluate the diagnostic relevance of pri-miR526b and pri-miR655 as biomarkers, we conducted ‘sensitivity’ and ‘specificity’ tests using Medcalc and GraphPad statistical software. We used sensitivity (detection of true disease or true positives) and specificity (ruling out the disease-free individuals or false positives) as measures of accuracy of pri-miRNAs as biomarkers [23,24]. We used the plot of sensitivity versus specificity known as the ROC curve and, for accuracy of the test, we used the area under the curve (AUC) [24]. An AUC close to 80% is considered a good sensitive test, with 100% indicating the best test to distinguish control and diseases.

To test the effectiveness of biomarkers in detecting cancer at an early stage, we compared the positive predictive value (PPV) and the negative predictive values (NPV) of both plasma pri-miRNAs at the earliest tumor stage. PPV is the probability that subjects with a positive screening test truly have the disease, and NPV is the probability that subjects with a negative screening test are truly free of disease. The accuracy of both pri-miRNAs was tested by first determining a cutoff point for each pri-miRNA on a normal distribution curve. Then, the accuracy for each pri-miRNA test was determined using MedCalc’s Diagnostic test Software.

### 2.6. Statistical Analysis

We used MedCalc statistical packages and GraphPad prism for statistical analysis. To evaluate the diagnostic performance of both pri-miRNAs, we used ROC and AUC analysis. Pri-miRNA as a risk factor for breast cancer was calculated using odds ratios (OR) and 95% confidence intervals (CI) using a chi-square test with Yates’s correction. For the comparison of proportions within the same sample set, we used *Z*-score analysis, and, to compare the means of two datasets, we performed a *t*-test. All parametric data were analyzed with one-way analysis of variance (ANOVA) by Tukey–Kramer or Dunnett post hoc comparisons. Statistically significant differences between means were accepted at *p* < 0.05.

## 3. Results

### 3.1. Clinicopathological Characteristics of Benign and Breast Cancer Patients

The blood plasma of the control subjects was collected from benign tumor-bearing individuals or nontumor lesions (control, *n* = 20) at the LTB. Control subjects were diagnosed as being cancer-free and having diverse histopathological lesions (Table 1). The lesions ranged from normal breast tissue to fibrosis, necrotic fat, adenomas of different types including fibrocystic, and papillary forms. Two of these control subjects later developed cancer 5 years post sample collection; thus, we did not exclude these two samples from further analysis.

For the LTB samples, the age of breast cancer and control subjects ranged from 42–91 years (mean age 64) and 52–87 years (mean age 66), respectively. Both cancer and control samples were age-adjusted, and there was no statistical difference in mean age distribution between the breast cancer and control subjects (Table 2). In the tumor dataset (case, *n* = 90), there were 81.11% estrogen receptor-positive (ER+ve), 75.56% progesterone receptor-positive (PR+ve), 78.89% human epidermal growth factor receptor 2 (HER2)-negative (HER2−ve), and only 8.89% triple-negative (ER−ve, PR−ve, and HER2−ve) receptor subtypes. The tumor staging data revealed that the majority of the tumors are stages I and II and all other subtypes are less than 13% (Table 2).

We used 20 breast cancer tissue samples (case) and 20 noncancerous tissues (control) from the OICR Tumor Bank. We previously reported this sample set and tested mature miRNA expression [14,16,17,25,26]. The demography of these samples is presented in Appendix A. All samples used here were from female participants, ages ranged between 27 and 87 years for control and between 32 and 92 years for breast cancer samples. Of the cancer samples, 75% were ER+ve, 14% were PR+ve, 60% were HER2−ve, and there were no triple-negative samples. Moreover, this set had the majority of stage II tumors (65%) followed by 25% stage III tumors, but there was no stage IV tumor. We only had one of each stage I sample and stage 0 sample (T = tumor size between 3 and 5 cm, pathological T value of T0; N = lymph node metastasis or pathological N of N0; M = pathological data for metastasis or clinical M of M0).

### 3.2. Levels of miRNA and Pri-miRNA Expressions Are Comparable in Breast Cancer

We measured the expressions of both mature miRNAs, miR526b and miR655 (normalized to RNU44), and pri-miRNAs, pri-miR526b and pri-miR655 (normalized to RPL5), in a small cohort of plasma and biopsy tissues (*n* = 15). This experiment was done to check the variability in the expression of miRNA and pri-miRNA within the same sample. There was no significant difference in the normalized miRNA and pri-miRNA expressions of miR526b (mean ± SEM: miR526b, 2.83 ± 1.37; pri-miR526b, −0.79 ± 1.20; *p* = 0.06) and miR655 (mean ± SEM: miR655, −1.29 ± 2.02; pri-miR655, −4.37 ± 0.62; *p* = 0.16) in malignant plasma samples (Figure 1A).

Next, we wanted to test the variability in the expression of the same miRNA and pri-miRNA in biopsy samples. We conducted a pairwise *t*-test in 15 matched biopsy samples, and the results showed no significant difference in the expression of both miR526b and miR655 with corresponding pri-miR526b and pri-miR655 in the tumor tissues for miR526b (mean ± SEM: 3.20 ± 0.46) and pri-miR526b (mean ± SEM: 2.03 ± 0.43) and for miR655 (mean ± SEM: −0.48 ± 0.93) and pri-miR655 (mean ± SEM: 1.17 ± 0.36; *p* = 0.11) (Figure 1B). The above results confirm that we can measure pri-miRNA instead of miRNA for screening plasma specimens. This would also exclude the need for enrichment methods required for miRNA quantification. Pri-miRNA can be extracted and measured using regular RNA extraction procedures and cDNA made with random primers. At the same time, the cDNA synthesized from total RNA could be used to measure many more cancer-related genes and markers along with pri-miRNA expression.

Next, we tested the variability of the endogenous control gene expression in plasma samples from control (benign) (*n* = 15) and BC subjects (*n* = 15). We selected *RPL5* as an endogenous gene and tested its stability in randomly selected plasma samples. Expression of RPL5 in the control samples had a Ct range of 29.97–30.94 and in BC plasma samples had a Ct range of 28.13–31.46 (Figure 1C). We compared the mean Ct values of RPL5 in both case (mean 29.11) and control (28.80) samples using a *t*-test. No significant variation in RPL5 expression between breast cancer and the control plasma samples was found (*p* = 0.5). Therefore, RPL5 is an ideal control gene to measure the relative expression of pri-miRNA in both sample sets.

### 3.3. Pri-miR526b and Pri-miR655 Expression in Various Tumor Stages of Plasma Samples

Pri-miRNA expressions were significantly high in BC plasmas (*n* = 90) compared to control plasmas (*n* = 20): pri-miR526b (mean ΔCt ± SEM, in BC 1.55 ± 0.49 vs. in control 5.10 ± 0.96) and pri-miR655 (mean ΔCt ± SEM, in BC −0.80 ± 0.40 vs. in control 1.28 ± 1.02), with *p*-values of 0.0023 and 0.0355 for pri-miR526b and pri-miR655, respectively (Figure 2A,B). Therefore, pri-miRNA expression is associated with breast cancer.

Next, we stratified tumor samples according to tumor stages and compared the expression of both pri-miRNAs across tumor stages. Pri-miR526b (mean ΔCt ± SEM, 1.46 ± 0.62; *p* = 0.0018) and pri-miR655 (mean ΔCt ± SEM, −0.95 ± 0.57; *p* = 0.0001) showed a significantly higher expression in the plasma of stage I tumors *n* = 43 compared to the benign controls (*n* = 20) (Figure 2C,D). Hence, the expression of both pri-miRNAs can distinguish early tumor stage vs. benign control.

We also compared stage I plasma pri-miRNA expression with both stage II and combined stages III and IV plasmas; however, stagewise comparisons were not significant.

### 3.4. Diagnostic Performance of Plasma Pri-mir526b and Pri-mir655 in Early Detection of BC

Both pri-miR526b and pri-miR655 showed high sensitivity values in detecting true disease in breast cancer plasma samples. Data showed that both pri-miR526b and pri-miR655 expressions in the plasma showed a high PPV of 88.89% with a confidence interval (CI) of 80.51–94.54 for pri-miR526b and 84.44% with a CI of 75.28–91.23 for pri-miR655. Plasma pri-miR526b showed an NPV of 35% and pri-miR655 showed an NPV of 10% (Table 3). Data also showed that both pri-miRNAs can detect disease with high accuracy values; pri-miR526b can predict cancer with 80.70% accuracy, while pri-miR655 can detect disease with 71.82% accuracy. Therefore, pri-miR526b is highly sensitive in predicting true disease.

We further conducted a risk and receiver operating curve analysis to evaluate the diagnostic performance of both pri-miRNA markers. Considering all tumor and control samples together, for a total of 110 samples, a cutoff or threshold ΔCt was decided. AUC is presented in Figure 3, and a summary of ROC analysis, AUC, 95% CI, and *p*-values in non-stratified samples is listed in Table 4. With a ΔCt cutoff value of 6.63 for miR526b, we found an AUC of 71.47% with a very significant *p*-value of 0.0027 (Figure 3A), with 86% sensitivity and a moderate specificity of 41% (Table 4). For miR655, the ΔCt cutoff was 3.63, which showed an AUC of 58.56% (Figure 3A), with a nonsignificant *p*-value, 80% sensitivity, and 12% specificity (Table 4).

In stratified tumor samples, we compared stage I tumor plasmas (*n* = 43) to control samples (*n* = 20); the AUC for miR526b increased to 72.73% (Figure 3B) with an increase in sensitivity and significant *p*-value of 0.0037. However, no significant increase in AUC was observed for miR655. Next, we compared the stage I tumor plasma to stage II (Figure 3C) and merged tumor stages III and IV (Figure 3D). However, the AUC was reduced to 53.13% and 55.37% for the stage I vs. II comparison for pri-miR526b and pri-miR655, respectively. Similarly, for the stage I vs. stages III and IV comparison, AUC was 59.09% and 57.61% for pri-miR526b and pri-miR655, respectively. A summary of the ROC analysis, AUC, 95% CI, and *p*-values is listed in Table 4.

For unstratified tumor samples, pri-miR526b showed a sensitivity of 86.02% with 95% CI of (77.28–92.34%) and a specificity of 41.18% with 95% CI (18.44–67.08%), which was very significant (*p* = 0.0027). In unstratified tumor samples vs. control, the sensitivity of pri-miR655 was 80.85% with 95% CI (71.44–88.24%), but this was not significant. Thus, pri-miR526b is a better and sensitive marker to detect cancer in the plasma. In stratified stage I tumor samples compared to control, for pri-miR526b, sensitivity was 75% with 95% CI (61.05–85.97), and this was very significant with a *p*-value of 0.0037 (Table 4). On the other hand, for pri-miR655, sensitivity reduced to 67% and specificity increased to 22%, but this was not significant. The AUC analysis was not significant for other stratified tumor stages compared to control. This could also have been because we had only a few samples with stages III and IV.

### 3.5. High Pri-miRNA Expression in Stratified Tumor Samples

High pri-miRNA expression was defined by a negative ΔCt value. The mean of ΔCt was considered as cutoff, and samples showing lower than the mean were considered as high-expression samples. The proportion of high pri-miRNA expression (negative ΔCt)/total number of tumors) was measured in various tumor stages and hormone statuses to identify if any group showed higher expression. The *Z*-score analysis to compare two proportions showed no difference in pri-miRNA expression among cancer stages (data not presented). However, we observed a significantly higher proportion of high-pri-miRNA-expression samples for the ER-positive and HER2-negative cancer subtypes (*p* < 0.04) (Table 5). Over 70% of LTB samples were from the ER/PR-positive and HER2-negative cancer-bearing subjects.

### 3.6. Pri-miRNA as a Risk Factor for Breast Cancer

We conducted a chi-square test comparing high miRNA expression (cutoff was decided on the basis of the mean ΔCt, where a value above the mean denoted a low expression and a value below the mean denoted a high expression) in case and control and calculated an odds ratio (OR). We found that pri-miR526b expression was significantly associated (*p*-value 0.01) with breast cancer, with an OR (95% CI) = 4.3 (1.39–13.34) (Table 6). However, the high expression of miR655 did not show an association with breast cancer. Here, we present a novel finding that high pri-miR526b expression in blood plasma is a risk factor for breast cancer onset.

### 3.7. Validation of Pri-miRNA Expression in OICR Samples

In selected tumor (*n* = 20) and control (*n* = 20) samples from the OICR tumor bank, we measured miRNA and pri-miRNA expressions. We previously reported high expression of miR526b [16] and miR655 [17] in breast tumors compared to disease-free control in this same sample set. Here, we measured the expression of pri-miR526b and pri-miR655 in the OICR same set. We tested the variability of miRNA and pri-miRNA expression in the same tumor sample; there was no significant difference in miR526b (mean ΔCt ± SEM, 2.69 ± 1.35) and pri-miR526b (mean ΔCt ± SEM, 6.71 ± 1.68) (Figure 4A) and in miR655 (mean ΔCt ± SEM, 1.42 ± 0.94) and pri-miR655 (mean ΔCt ± SEM, 2.49 ± 1.32) (Figure 4B) expressions. In comparing tumor to control tissue expression, pri-miR526b (mean ΔCt ± SEM for control 16.55 ± 2.09 and for case 6.71 ± 1.68, *p* = 0.0007) (Figure 4C) and pri-miR655 (mean ΔCt ± SEM for control 11.45 ± 1.18 and for case 2.49 ± 1.32, *p* = 0.0001) (Figure 4D) expression was significantly high in tumors. These results show that pri-miRNAs are robust markers, and expression of pri-miRNA can be found in various tumor tissues. There was no blood plasma sample available in the OICR tumor bank; thus, we were unable to find pri-miR526b or pri-miR655 expression. Moreover, we were unable to find any other report on pri-miRNA expression in the blood of breast cancer patients.

## 4. Discussion

In Canada and other countries around the globe, breast cancer incidence in younger women continues to rise. Mammographic screening starts at the age of 50, and this procedure contributes to 9% of all false-positive results [25]. However, if cancer is detected early, a patient’s survival is enhanced up to 98% [26]. This highlights the necessity of identifying a blood-based biomarker that would detect BC at an early stage with a minimally invasive blood test. Recently, circulating markers in the blood such as miRNAs and pri-miRNAs have emerged as highly sensitive biomolecules, which can be used as diagnostic and prognostic biomarkers [7,13].

We have established that mature miRNAs, miR526b and miR655, are oncogenic and tumor-promoting in breast cancer, and miRNA functions are regulated by the COX-2/EP4/PI3K/Akt signaling pathways [16,17,19,20,21]. We have also shown that these two miRNAs can be found in the cell-free conditioned media of miRNA-overexpressing cell lines [19] and these miRNAs can change the TME and enhance tumor-associated angiogenesis [20], hypoxia [21], and oxidative stress [19] in breast cancer. Extracellular vesicles carry miRNAs and release them into the interstitial fluid, while cell-free miRNAs have also been found in blood plasma and are being tested as blood-based biomarkers [7,27,28,29]. However, specific miRNA detection in the blood plasma has some limitations, including the necessity of a large volume of specimens for miRNA-specific cDNA synthesis and detection. This also limits the capacity to screen a panel of miRNAs per sample. To resolve these issues and to find a reliable and sensitive biomarker, in this study, we tested the diagnostic potential of pri-miR526b and pri-miR655 to serve as blood biomarkers in breast cancer.

We could measure pri-miRNA in the blood plasma using standard total RNA extraction and gene expression procedures, thereby reducing the demand for a large quantity of patient specimens, while pri-miRNAs are also highly sensitive in the detection of disease. We found that expressions of both pri-miR526b and pri-miR655 in the plasma of cancer patients were significantly high relative to the control. There are no pri-miRNA expression data available in The Cancer Genome Atlas (TCGA), and we were unable to find any report on pri-miRNA as a blood biomarker for breast cancer. We could only find one report of circulating mature miRNA in the plasma as an early detection biomarker in breast cancer [11]. Therefore, we are the first to report pri-miRNA as a breast cancer biomarker.

We also observed that both pri-miR526b and pri-miR655 expression in the plasma can distinguish benign lesions from tumor stage I. This is a very significant finding; this indicates the potential of pri-miRNA to serve as an early diagnostic biomarker. As reported by another group, the systemic plasma miRNA expression was heightened in breast cancer patients at various tumor stages [30]. Hence, to examine the varying expressions of pri-RNAs in plasma as an indicator of tumor progression, we compared pri-miRNA expression across tumor stages. However, we did not find a significant difference between stage I and Higher-stage tumors. This could have been due to few samples of higher-stage tumors (stage III and IV) in our dataset.

In stratified samples, we conducted a *Z*-score analysis to determine if there was any significant difference in the proportional distribution of high pri-miRNA expression. We found the distribution of pri-miR526b and pri-miR655 in plasma of ER-positive cancer patients to be significantly higher compared to that in the ER-negative cancer patients. We also observed a significantly higher proportion of high pri-miRNA expression in HER2-negative compared to the HER2-positive tumor plasma. This is very crucial since ER-positive and HER2-negative breast cancers are the predominant subtypes of breast cancer detected worldwide [2,3]. Therefore, these two pri-miRNAs can be used as biomarkers to distinguish ER-positive from ER-negative tumors and can be used to manage endocrine therapy. This is supported by another study showing similar effects to miRNA expression in distinguishing hormone receptor-positive vs. -negative tumors [31].

Next, we tested the sensitivity and specificity of pri-miRNA to serve as blood biomarkers, and we found that both pri-miRNAs can distinguish metastatic vs. benign tumors, with pri-miR526b standing out as a highly sensitive biomarker. Although pri-miR655 has a much lower false-negative rate (13%), the expression of pri-miR526b had an optimum sensitivity score of 86.02%, representing the strongest evidence to serve as a biomarker. Pri-miR526b also showed high sensitivity to distinguish stage I plasma from control with a sensitivity score of 75%. However, pri-miR526b expression did not show significant sensitivity to distinguish stage I tumor plasma from late-stage subjects. Thus, pri-miR526b has potential to serve as a diagnostic biomarker in breast cancer.

Furthermore, we tested if the high expression of these two pri-miRNAs in the blood is a risk factor for breast cancer. There are a few reports showing polymorphisms in pre-miRNA gene or miRNA expression as a risk factor associated with breast cancer [32,33]. However, to our knowledge, this is the first time that pri-miRNA expression in blood plasma has been directly established as a risk factor for breast cancer. Here, we report that pri-miR526b expression is a risk factor in breast cancer. High pri-miR526b expression in plasma is associated with breast cancer (*p* = 0.01) with an odds ratio of 4.3 and confidence interval (CI) = 1.391–13.34. However, this observation needs to be validated in a larger sample set.

To show the robustness of pri-miRNA expression, we tested both pri-miRNA expressions in another set of tumors and control tissues from the OICR tumor bank. We could not find any significant difference in mature and pre-mature miRNA expression within the same tissue. However, we found that both pri-miRNAs were significantly high in tumor samples compared to control. These results further strengthen the role of pri-miR526b and pri-miR655 as robust breast cancer markers.

Mature miR526b and miR655 expressions are regulated by COX2 and EP4, and inhibition of COX2 and EP4 with a specific COX-2 inhibitor and EP4-antagonist (EP4A) significantly inhibited miRNA-induced functions in breast cancer [23,24,25]. Moreover, EP4A has already been approved by the FDA for arthritis treatment [34]. Therefore, the establishment of a link between these pri-miRNA expressions with COX-2 and EP4 expressions in blood plasma would be interesting. Hence, in the future, these two pri-miRNAs could be used as screening tools for monitoring EP4A treatments.

## 5. Conclusions

Circulating pri-miR526b and pri-miR655 in the blood can improve noninvasive BC detection and can serve as liquid biopsy alongside traditional BC screening procedures. Pri-miR526b can serve as an early diagnostic biomarker for BC. In ER-positive breast cancer, these pri-miRNAs might play a role in the management of endocrine therapy. Precursors of miRNA could be considered as a novel breast cancer blood biomarker.

## Figures and Tables

**Figure 1 cancers-13-03838-f001:**
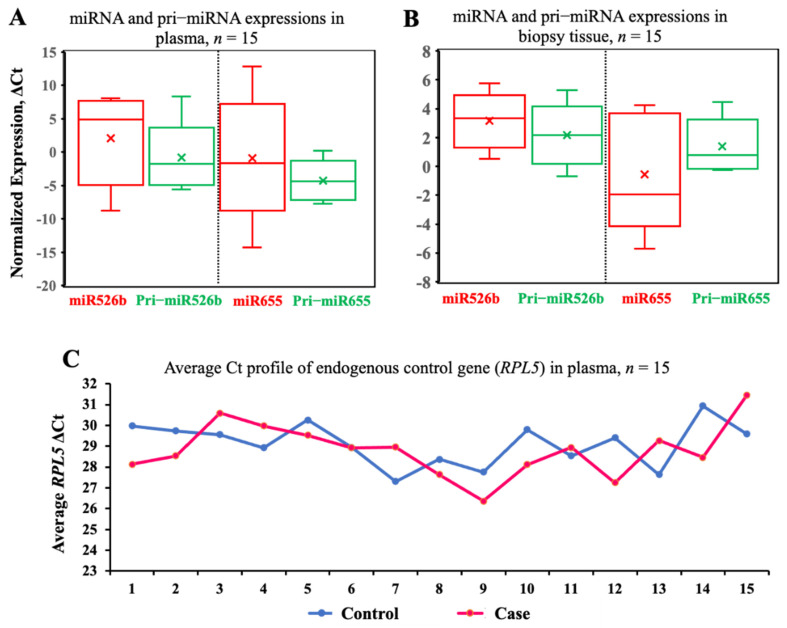
Levels of miRNA and pri-miRNA expressions are comparable in breast cancer specimens. (**A**,**B**) Boxes show pairwise analysis of normalized expressions of both mature miRNA and pri-miRNA in breast cancer patients’ plasma and matched cancer tissue samples from the same BC subjects (*n* = 15). Comparisons in A and B are not statistically significant. (**C**) Scatter plot of RPL5 expression in control (*n* = 15) and breast cancer (*n* = 15) patient plasma. This comparison was not statistically significant.

**Figure 2 cancers-13-03838-f002:**
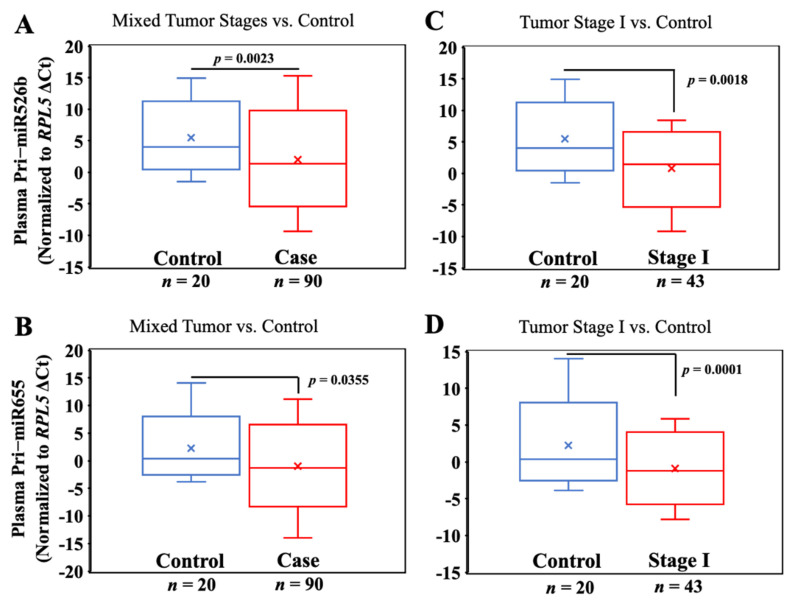
Plasma pri-miR526b and pri-miR655 expressions are high in BC. (**A**,**B**) Box plots showing expression of pri-miR526b and pri-miR655 in plasma of benign (control) vs. breast cancer subjects (unstratified); (**C**,**D**) benign (control) vs. stage I cancer.

**Figure 3 cancers-13-03838-f003:**
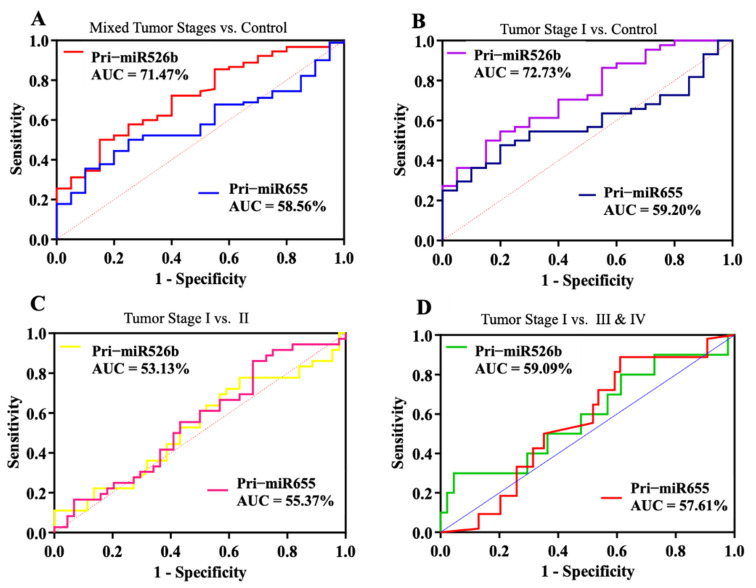
Diagnostic performance of pri-miR526b and pri-miR655. Receiver operating characteristic (ROC) plots show pri-miR526b and pri-miR655 diagnostic test curves with AUC for (**A**) unstratified cancer samples vs. control (benign), (**B**) stage I vs. control, (**C**) stage I tumor vs. stage II, and (**D**) stage I tumor vs. late cancer stages (III and IV).

**Figure 4 cancers-13-03838-f004:**
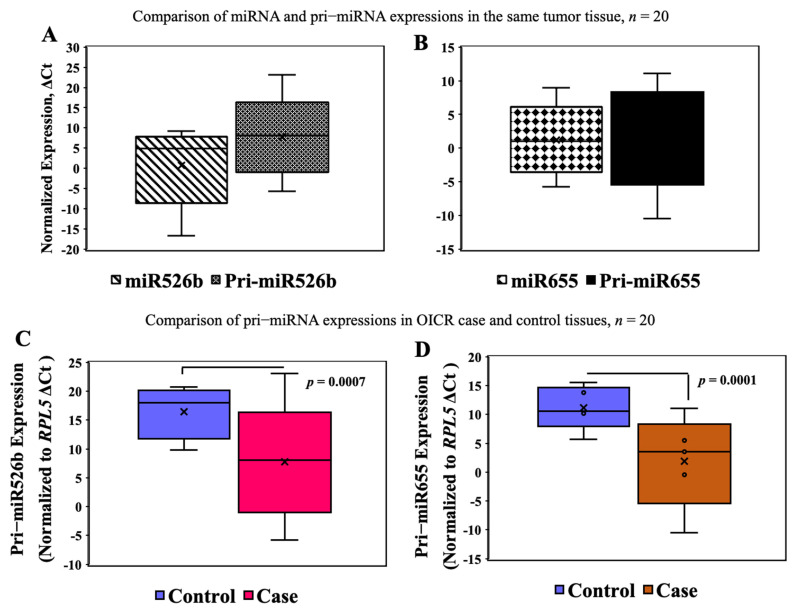
miRNA and pri-miRNA expression in the OICR case (*n* = 20) and control (*n* = 20) tissues. (**A**,**B**) There was no significant difference between miRNA and pri-miRNA expression in the same tumor samples. (**C**,**D**) Expression of pri-miR526b and pri-miR655 was significantly high in OICR tumor tissues (case) compared to control.

**Table 1 cancers-13-03838-t001:** Histopathology and demographics of benign subjects.

Sample No.	Pathology	Histopathological Details	Age	Diagnosed with Cancer after Specimen Collection
C1	Benign	Atypical ductal hyperplasia within a complex, sclerosed papilloma; fibroadenoma	59	No
C2	Benign	Fibrosis intermixed with foamy macrophages and numerous stromal microcalcifications, suggestive of healing fat necrosis	90	No
C3	Benign	Benign phyllodes tumor	51	No
C4	Benign	The lesion is a benign fibroadenoma, but the patient has a recent history of breast cancer within the past 6 months; undergoing adjuvant treatment at the time of this biopsy	55	No
C5	Benign	Fibroadenoma with focal with florid ductal hyperplasia	52	No
C6	Benign	Focal cyst rupture, with associated chronic inflammation, fibrosis, collections of foamy histiocytes, and multinucleated giant cell reaction	59	Yes, in 2018
C7	Benign	Papillary lesion without atypia	82	No
C8	Benign	Nonproliferative change	50	No
C9	Benign	Fibrocystic and fibro adenomatoid changes, focal epithelial hyperplasia	51	No
C10	Benign	Fragments of reactive lymph node tissue, unremarkable breast parenchyma	61	No
C11	Benign	Biopsy specimen: nonproliferative (fibrocystic) change (fibro adenomatoid hyperplasia, cysts, fibrosis); lumpectomy specimen: proliferative breast disease without atypia (moderate ductal epithelial hyperplasia, columnar cell type); regional dense stromal fibrosis	44	No
C12	Benign	Papilloma	60	No
C13	Benign	Benign fibro glandular tissue with focus of inflammation suggestive of resolving fat necrosis	64	No
C14	Benign	Focal abscess with mixed acute and chronic inflammation, histiocytes, and fibrosis	36	No
C15	Benign	Atypical ductal hyperplasia (ADH) involving a complex intraductal papilloma	46	Yes, in 2017
C16	Benign	Sclerosed complex papillary lesion	57	No
C17	Benign	Fibroadenoma	46	No
C18	Benign	Fat necrosis with calcifications, florid ductal epithelial hyperplasia	48	No
C19	Benign	Smooth muscle tumor	63	No
C20	Benign	Complex papillary lesion without atypia	59	No

**Table 2 cancers-13-03838-t002:** Clinical characteristics of breast cancer subject and control subject demographics.

Characteristics		Control *n* = 20	Breast Cancer *n* = 90 (100%)
Sex	Male	0	0
	Female	20	90
Age (years)	Mean ± SD (range)	66 ± 11 (52–87)	64 ± 12 (42–91)
Tumor Receptor Status	ER+ve (%)		73 (81.11)
	ER−ve (%)		17 (18.89)
	PR+ve (%)		68 (75.56)
	PR−ve (%)		22 (24.44)
	HER2+ve (%)		19 (21.11)
	HER2−ve (%)		71 (78.89)
	ER/PR/HER2−ve (%)		8 (8.89)
TNM (Tumor) Staging	0		1 (1.11)
	I		43 (47.78)
	II		36 (40.00)
	III		8 (8.89)
	IV		2 (2.22)

**Table 3 cancers-13-03838-t003:** Test for positive predictive value (PPV) and the negative predictive values (NPV) and accuracy. Both pri-miRNAs are highly sensitive (>80%) in detecting disease with high accuracy (>71%). Pri-miR526b showed higher PPV and NPV values compared to pri-miR655.

	Mean Ct Cut-Off	Sensitivity (%)	Specificity (%)	PPV (95% CI)	NPV (95% CI)	Accuracy (95% CI)
Pri-miR526b	6.73	86.02	41.18	88.89 (80.51–94.54)	35.00 (15.39–59.22)	80.70 (71.30–87.02)
Pri-miR655	3.63	80.85	12.5	84.44 (75.28–91.23)	10.00 (1.24–31.70)	71.82 (62.44–79.98)

**Table 4 cancers-13-03838-t004:** Test for significance of AUC analysis in BC and benign plasma samples. In unstratified samples, pri-miR526b showed an AUC of 71.47% with a significant *p*-value. However, for pri-miR655, the AUC was not significant. In stratified samples, pri-miR526b showed a sensitivity of 75% in distinguishing stage I tumors from benign samples with a significant *p*-value of 0.0037, whereas pri-miR655 did not show any significant AUC.

	Unstratified Tumor vs. Control		Stratified Tumor (Stage I) vs. Control	
	Pri-miR526b	Pri-miR655	Pri-miR526b	Pri-miR655
AUC (95% CI)	71.47 (59.80–83.14)	58.56 (46.86–70.25)	72.73 (59.84–85.62)	59.20 (45.40–73.01)
*p*-value	0.0027	0.2327	0.0037	0.2408
Sensitivity (95% CI)	86.02 (77.28–92.34)	80.85 (71.44–88.24)	75.00 (61.05–85.97)	67.27 (53.29–79.32)
Specificity (95% CI)	41.18 (18.44–67.08)	12.50 (1.55–38.35)	58.33 (27.67–84.83)	22.22 (2.815–60.10)

**Table 5 cancers-13-03838-t005:** High expressions of plasma pri-miRNAs in stratified tumor samples. Table showing the distribution of high pri-miR526b (lower than mean ΔCt 6.73 is high) and high pri-miR655 (lower than mean ΔCt 3.63 is high) expression in the BC subjects according to ER/PR/HER2 tumor status. The Z-score analysis demonstrated a significantly higher proportion of ER+ve and HER2−ve tumor plasma samples showing high expression of pri-miR526b and pri-miR655.

Tumor Status	No. of Samples *n* (%)	High Pri-miR526b *n* (%)	Z-Score	*p*-Value	High Pri-miR655 *n* (%)	Z-Score	*p*-Value
ER+ve	73 (81.1)	60 (82.1)	2.0872	0.0366	58 (79.4)	2.7201	0.0065
ER−ve	17 (18.9)	10 (58.8)			8 (47.0)		
PR+ve	68 (75.6)	55 (80.9)	1.2455	0.2113	53 (77.9)	1.7379	0.0819
PR−ve	22 (24.4)	15 (68.2)			13 (59.1)		
HER2+ve	19 (21.1)	10 (52.6)	−2.7888	0.0053	10 (52.6)	−2.1465	0.0316
HER2−ve	71 (78.9)	59 (83.1)			55 (77.5)		

**Table 6 cancers-13-03838-t006:** Pri-miRNA expression is associated with breast cancer. With the given ΔCt cutoff value of 6.73 (mean of all ΔCt), pri-miR526b expression was significantly high in cancer samples compared to control and was significantly associated with breast cancer. However, for high pri-miR655 expression with a ΔCt cutoff of 3.63, we did not find any significant difference between benign and malignant plasma.

	Ct Cutoff	BC High Pri-miRNA	Benign High Pri-miRNA	Odds Ratio	95% CI	*p*-Value
Pri-miR526b	6.73	80	13	4.308	1.391–13.34	0.0142
Pri-miR655	3.63	76	18	0.6032	0.126–2.895	0.7316

## Data Availability

All data related to this publication is available in the article and additional data can be found in the Appendix A.

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
