# Peer review of "Pri-miR526b and Pri-miR655 Are Potential Blood Biomarkers for Breast Cancer"

_cancers, 2021, doi:10.3390/cancers13153838_

Round 1
Reviewer 1 Report
The authors have revised manuscript according to the reviewer's suggestions. The manuscript is suitable for publication in the journal "Cancers."
Reviewer 2 Report
no
This manuscript is a resubmission of an earlier submission. The following is a list of the peer review reports and author responses from that submission.
Round 1
Reviewer 1 Report
The manuscript by Ugwuagbo et al. entitled “ Pri-miRNA-526b and -655 as potential blood biomarkers for breast cancer” reports the utility of Pri-miRNA-526b and -655 in the early detection of breast cancer. The pilot study data presented in this manuscript is highly significant in view of the detectability of Pri-miR526b with high sensitivity and accuracy in plasma samples. the study performed is scientifically sound and appropriate statistical analyses was performed.
The functional relevance of miRNAs in gene regulation in oncology paves the way for the in-depth understanding of oncogenesis. The relevance of pre-miRNAs as biomarkers in breast cancer is interesting and would appeal to a larger audience of the journal “Cancers.”
Major Comments:
Methodology: The authors should indicate the type of study in the method section. If this is a case-control cohort study, then that should be clarified in the method section. A rationale for different housekeeping genes used for the miRNA and pri-miRNAs should be included in the method section. The authors should explain patient stratification in the method section.
Results and Discussion: The AUC values reported for various comparative parameters show different values. It is not clear how the accuracy of 81% was achieved. The authors should clarify the accuracy calculation. The inclusion of a table showing various cutoff points for Ct with sensitivity and specificity would provide a clear depiction of accuracy. Although the expression data of both pri-miRNAs show significant expression in BC compared to controls, the diagnostic evaluation indicates that the pri-miR526b is a robust marker in predicting BC, this aspect should be emphasized in the abstract and discussion. The results indicate that both pri-miR526b and pri-miR655 are significantly overexpressed in ER+ and Her2Nue BC. Since, majority of BC are ER/PR positive, these results are highly significant in the management of ER positive BC endocrine therapy designs. The significant differential expression of pri-miR526b and pri-miR655 in ER+ and HER2 negative but not significant in PR positive is intriguing. The authors should discuss the potential reason for pri-miR526b and pri-mi655 significantly overexpressed in ER+ BC in the discussion section.
Conclusion: the authors should add the significance of pri-miR526b as a molecular marker for the detection of ER+ BC because of the potential translational aspect in the preliminary detection of ER+ BC and endocrine therapy design.
Minor comments:
The authors should take care of typos.
Author Response
Comment: The manuscript by Ugwuagbo et al. entitled “Pri-miRNA-526b and -655 as potential blood biomarkers for breast cancer” reports the utility of Pri-miRNA-526b and -655 in the early detection of breast cancer. The pilot study data presented in this manuscript is highly significant in view of the detectability of Pri-miR526b with high sensitivity and accuracy in plasma samples. The study performed is scientifically sound and appropriate statistical analyses were performed.
The functional relevance of miRNAs in gene regulation in oncology paves the way for the in-depth understanding of oncogenesis. The relevance of pri-miRNAs as biomarkers in breast cancer is interesting and would appeal to a larger audience of the journal “Cancers.”
Response: Thank you so much for your very encouraging comments and reviews. Your comments improved the quality of this article manyfold. We have now addressed every major or minor concern you had. Please note all revisions in the manuscript are highlighted.
Major Comments:
Methodology: The authors should indicate the type of study in the method section. If this is a case-control cohort study, then that should be clarified in the method section. A rationale for different housekeeping genes used for the miRNA and pri-miRNAs should be included in the method section. The authors should explain patient stratification in the method section.
Response: Please see the point-by-point response below.
Comment 1A: Authors should indicate the type of study in the method section
Response: This is an excellent comment. We have now revised the method section and clearly stated that this is a pilot biomarker study. In addition, we have also conducted a case-control association study and calculated the odds ratio and confidence interval. Please see the method section, line number 103-129.
Comment 1B: Authors should state the rationale for using RNU44 and RPL5 reference genes in the method section
Response: Thank you. We now have explained this in the manuscript. According to Majumder et al (2015 and 2018), authors have shown RNU44 to be a stable control marker for miRNA expression in both cell lines and breast tissues. Also, in Hutner et al (2019), Shin et al (2019) and Gervin et al (2020), the authors have shown RPL5 as an ideal endogenous control gene that showed consistent expression in both cell lines and breast tumor and control tissues. In this current article, we have also shown that RPL5 expression did not vary significantly from sample to sample. Please see lines 159-163 in the method section.
Comment 1C: Authors should explain patient stratification in the method section
Response: Study subject stratification has been addressed in the method section 2.1. Study design, patient demography, and study subjects. Please note the changes in lines 106-129.
Comment
Results and Discussion: The AUC values reported for various comparative parameters show different values. It is not clear how the accuracy of 81% was achieved. The authors should clarify the accuracy calculation. The inclusion of a table showing various cutoff points for Ct with sensitivity and specificity would provide a clear depiction of accuracy. Although the expression data of both pri-miRNAs show significant expression in BC compared to controls, the diagnostic evaluation indicates that the pri-miR526b is a robust marker in predicting BC, this aspect should be emphasized in the abstract and discussion. The results indicate that both pri-miR526b and pri-miR655 are significantly overexpressed in ER+ and Her2Nue BC. Since the majority of BC are ER/PR positive, these results are highly significant in the management of ER-positive BC endocrine therapy designs. The significant differential expression of pri-miR526b and pri-miR655 in ER+ and HER2 negative but not significant in PR positive is intriguing. The authors should discuss the potential reason for pri-miR526b and pri-mi655 significantly overexpressed in ER+ BC in the discussion section.
Response: Thank you so much for helping us to improve the quality of this work. Please see below the point-by-point revision that we have made in the manuscript.
Comment 2A: Authors should explain how the accuracy of 81% was achieved and include a table showing the cut-off points for Ct with sensitivity and specificity
Response: The accuracy of both pri-miRNAs was tested by first determining a cut-off point for each pri-miRNA on a normal distribution curve in a sample set. Then, calculating the true positive, true negative, false positive, and false negative values from each cut-off, the accuracy for each pri-miRNA test was determined using MedCalc's Diagnostic test Software. We added a separate accuracy value for each pri-miRNA on table 3 and clarified it in the results section, lines 302-309.
Comment 2B: Authors should clearly state (in Abstracts and Discussion section) that ‘Pri-miR526b is a robust marker in predicting Breast cancer’?
Response: This is an excellent suggestion. Thank you.
Now, we have added this in the abstract and discussion, lines 474-479. We have measured pri-miRNA expression in an additional set of samples from the OICR tumor bank. New data is provided in figure 4. In this independent data set, pri-miRNA expression reflects the same observation as we found with London Tumor Bank samples, both pri-miRNAs are high in breast tumors compared to control. However, OICR samples are only breast biopsy tissues, we do not have blood plasma for this set. Nevertheless, two independent data set results to show the pri-miRNAs are robust biomarkers in breast cancer, line 380-399.
Comment 2C: The results indicate that both pri-miR526b and pri-miR655 are significantly overexpressed in ER+ and Her2Nue BC. Since the majority of BC are ER/PR positive, these results are highly significant in the management of ER-positive BC endocrine therapy designs. The significant differential expression of pri-miR526b and pri-miR655 in ER+ and HER2 negative but not significant in PR positive is intriguing. The authors should discuss the potential reason for pri-miR526b and pri-mi655 significantly overexpressed in ER+ BC in the discussion section.
Response: Thank you for making this article more relevant and impactful.
Worldwide, most breast cancer patients are hormone receptor (ER/PR) positive and HER2- negative. The London Tumor Biobank sample set we used here is a true reflection of the same observation above. Hence, these results will play a major role in the endocrine therapy management for hormone receptor-positive breast cancer patients. We have now added this in the conclusion (line number 496-498) and highlighted it in the abstract.
Comments in Conclusion: the authors should add the significance of pri-miR526b as a molecular marker for the detection of ER+ BC because of the potential translational aspect in the preliminary detection of ER+ BC and endocrine therapy design.
Response: Authors have emphasized the significance of pri-miR526b as a marker for ER-positive BC detection in the CONCLUSION. Thank you so much.
Minor comments:
The authors should take care of typos.
Response: We have now checked for typos and corrected all.
Reviewer 2 Report
This study investigated the potenial role of two microRNAs for breast cancer diagnostics. Although the topic is innovative in biomarker research the trial design has some relevant limitations.
The results from the training group were not verified in an independent test group which is now standard in biomarker development.
10% of interval carcinomas within the follow up (recruiting period is not given) is an significant limitation and might be related to undetected initial malignant lesions. This has intensive influence on the ROC.
Does 44 patients had stage 0/1 in table 2 mean that premalignant lesions were included?
Validation of expression blood/tissue in only 5 patients is too small.
What about stage 2 in figure 4? Why were different y-axies were used in this figure making it difficult to compare the results. C/E and D/F need to be combined in one diagram since they directly belong together.
Dividing the relative small group of patients into subgroups (table 6) is, at least, critical.
The number of tables needs to be reduced for better readibility.
Author Response
Comment: This study investigated the potential role of two microRNAs for breast cancer diagnostics. Although the topic is innovative in biomarker research the trial design has some relevant limitations.
Response: Thank you for helping us to improve the quality of the article. We have now addressed every concern you raised. All changes are highlighted in the revised manuscript. Please see the point-by-point responses below.
Comment: The results from the training group were not verified in an independent test group which is now standard in biomarker development.
Response: This is excellent advice. We could not find any other report on pri-miRNA in blood plasma. Also, we looked further into the TCGA database, but we were unable to find any data on pri-miRNA expression in the blood plasma or tissues of breast cancer subjects. The only available data was on mature miRNA expressions in breast cancer. However, we have previously published that these two miRNAs, miR526b (Majumder M et al, Molecular Cancer Research, 2015) and miR655 (Scientific Reports, Majumder M, et al, 2018) are associated with breast cancer. So, we have tested pri-miRNA expression in independent tumor bank samples (OICR tumor bank). New data provided in figure 4. Accordingly, the demographic data is provided in supplementary table 1, while results and discussion are added in the revised manuscript, all highlighted in blue color.
Comment: 10% of interval carcinomas within the follow-up (the recruiting period is not given) is a significant limitation and might be related to undetected initial malignant lesions. This has an intensive influence on the ROC.
Response: Thank you.
We suppose you meant “10% of interval carcinomas” as 2 out of 20 benign samples later had cancer. These two samples were detected with the disease after five years of sample collection (samples were collected in 2011-2013 and diseases reported in 2017 and 2018 respectively). However, as you suggested, we have excluded these two samples from benign and recalculated the ROC, however that did not significantly change the AUC plots. So, we decided not to exclude these two samples from the control.
Comment: Do 44 patients who had stage 0/1 in table 2 mean that premalignant lesions were included?
Response: Thank you. No, we did not include any pre-malignant lesions under stage 0 or I. Now we showed the stage-wise distribution of tumors in revised table 2 (line 213-217).
Comment: Validation of expression blood/tissue in only 5 patients is too small.
Response: Thank you. Now we have shown data for 15 matched blood and tissue samples. Please see revised figure 1 (line 264-269).
Comment: What about stage 2 in figure 4? Why were different y-axes used in this figure making it difficult to compare the results? C/E and D/F need to be combined in one diagram since they directly belong together. Dividing the relative small group of patients into subgroups (table 6) is, at least, critical.
Response: Thank you for, excellent suggestions. Now we have provided AUC data for stage II in new figure 3C. We have also revised the axis and merged C with E and D with F. Please see the revised figure 3 (line 320-324).
Comment: The number of tables needs to be reduced for better readability.
Response: Thank you. We have reduced the number of tables. Now tables 3 and 4 are merged as table 3. However, the number of figures increased as we have conducted additional experiments and validated pri-miRNA expression in breast cancer in an independent data set (Figure 4). We have now additional data in supplementary table 1.
Reviewer 3 Report
Very interesting work with great potential for use in the near future in a breast cancer clinic. Application of liquid biopsy with Pri-miRNA-526b and -655 in combined diagnostic procedures. Additionally, it can be very useful in women under 50, especially in those using hormone replacement therapy and / or with a positive family history (BRCA 1, 2). As well as in women after breast cancer treatment in the framework of follow-up examination.
48 - How painful on a scale of 1-10?
- -high false-positive outcome rate, repeated in the discussion
Introduction - too extensive and detailed. The reader should be clearly and briefly introduced to the research topic. End points should be clearly and concisely defined (redundant especially 91-104).
107-109 I suggest at the end of the chapter
109 When the tests were performed
113 What type of surgical biopsy was performed (e.g., surgical excision, mammotomy biopsy, Tru-cat)
152-154 Redundant definition, generally known
178 - It is not necessary to repeat exactly all the data contained in the table
187- It is not necessary to repeat all the data in the table
171-344 In the results section we present only the obtained data without interpretation and research methods. Some of the additional data from these sections can be described in other places of work, e.g. in a discussion
344-361 Repetition (introduction), this does not apply to the current results
419-425 I propose to specify the conclusions!. 2., 3.
Author Response
Comment: Very interesting work with great potential for use shortly in a breast cancer clinic. Application of liquid biopsy with Pri-miRNA-526b and -655 in combined diagnostic procedures. Additionally, it can be very useful in women under 50, especially in those using hormone replacement therapy and/or with a positive family history (BRCA 1, 2). As well as in women after breast cancer treatment in the framework of follow-up examination.
Response: Thank you very much for your encouraging comments. We have done all the corrections as you suggested. All changes are highlighted in the revised article.
Comment: 48 - How painful on a scale of 1-10?
Response: Thank you. We understand your concern. Unfortunately, we were unable to find the measurable scale to compare mammogram pain scale. In the context of blood biomarkers which is a less invasive and less painful procedure, we compared liquid biopsy with a mammogram.
Comment: 50. -high false-positive outcome rate, repeated in the discussion
Response: Thank you. We now have removed the repeated content.
Comment: Introduction - too extensive and detailed. The reader should be clearly and briefly introduced to the research topic. End points should be clearly and concisely defined (redundant especially 91-104).
Response: Thank you, we reduced the redundant content and now wrote the introduction precisely.
Comment: 107-109 I suggest at the end of the chapter
Response: Thank you. Now this section is moved to the conclusion.
Comment: 109 When the tests were performed
Response: All experiments were performed at Brandon University between 2017-2021.
Comment: 113 What type of surgical biopsy was performed (e.g., surgical excision, mammotome biopsy, Tru-cat)
Response: Thank you for asking this. These are all core needle biopsy samples.
Comment: 152-154 Redundant definition, generally known
Response: Yes, thank you for this comment. However, to respond to one of the reviewer’s comments, we decided to keep these definitions.
Comment: 178 - It is not necessary to repeat exactly all the data contained in the table
Response: Thank you. We now have removed the repeated content.
Comment: 187- It is not necessary to repeat all the data in the table.
Response: Thank you. We now have explained the necessary content of the table briefly in the content.
Comment: 171-344 In the results section we present only the obtained data without interpretation and research methods. Some of the additional data from these sections can be described in other places of work, e.g. in a discussion.
Response: Yes, we agree with your point. Now, we have moved all definitions from results to methods and discussion.
Comment: 344-361 Repetition (introduction), this does not apply to the current results
Response: Yes, we agree. We re-written this section. Please see the revised discussion.
Comment: 419-425 I propose to specify the conclusions!. 2., 3.
Response: Thank you. Now, we have changed the conclusion. Please see the revised conclusion.
Round 2
Reviewer 2 Report
The data in table 2 and text at page 6 are not consistent.
Stage 0 is still not clear - this menas no cancer!
Looking at figure 2 it is impossible to belief all the significant differences, esp. in B), E) and F)
The veryy low specificitc is not acceptable for a screening method. Therefore, the conclusion is not supported by the data presented in the manuscript.
Author Response
Dear reviewer,
Thank you very much for your tedious revisions. We find the quality of the manuscript improved tremendously with your critical reviews. Please see the new edits highlighted as green in the manuscript.
Comment: The data in table 2 and the text on page 6 are not consistent.
Answer: We apologies for this mistake. Now we have corrected the percentage in Table 2. Please see page 6 for the revised table.
Comment: Stage 0 is still not clear - this means no cancer!
Answer: Thank you. As confirmed by the pathologist, the following is the parameter that was used to define stage 0. Please see page 7 in the manuscript.
“T= tumor size between 2-4cm, sometimes referred as carcinoma in situ, pathological T value is T0; N= lymph node metastasis or pathological N is N0; M= pathological data for metastasis or clinical M is M0”
Comment: Looking at figure 2 it is impossible to believe all the significant differences, esp. in B), E) and F)
Answer: Thank you. Please see revised figure 2. We have now provided p values on the figure and removed all marginally significant data. Please see page 9 for the revised figure 2 and accordingly, we have revised the result and discussion (highlighted in green).
Comment: The very low specificity is not acceptable for a screening method. Therefore, the conclusion is not supported by the data presented in the manuscript.
Answer: Please see the changed conclusion. Hope now this is more reflective of the results we found in this pilot study. We also rewrote sections in the discussion, abstract, and conclusion mentioning the limitations of the study and only presenting significant results. Please see all changes highlighted in green.
Also, we have done another round of english check for typo. Thank you.